# Interventions to improve perinatal outcomes among migrant women in high-income countries: a systematic review protocol

Kerrie Stevenson [iD],[1] K Ogunlana,[1] Samuel Edwards,[2] William G Henderson,[3] Hannah Rayment-Jones [iD],[4] Majel McGranahan,[2] Maria Marti-Castaner,[5] Gracia Fellmeth,[6] Serena Luchenski,[7] Fiona A Stevenson,[8] Marian Knight,[6] Robert W Aldridge [iD][1]

For numbered affiliations see end of article.

**Correspondence to**
Dr Kerrie Stevenson;
k.stevenson@ucl.ac.uk

## ABSTRACT

**Introduction** Women who are migrants and who are pregnant or postpartum are at high risk of poorer perinatal outcomes compared with host country populations due to experiencing numerous additional stressors including social exclusion and language barriers. High-income countries (HICs) host many migrants, including forced migrants who may face additional challenges in the peripartum period. Although HICs' maternity care systems are often well developed, they are not routinely tailored to the needs of migrant women. The primary objective will be to determine what interventions exist to improve perinatal outcomes for migrant women in HICs. The secondary objective will be to explore the effectiveness of these interventions by exploring the impact on perinatal outcomes. The main outcomes of interest will be rates of preterm birth, birth weight, and number of antenatal or postnatal appointments attended.

**Methods and analysis** This protocol follows the Preferred Reporting Items for Systematic Reviews and Meta-Analyses (PRISMA) Protocols guidelines. EMBASE, EMCARE, MEDLINE and PsycINFO, CENTRAL, Scopus, CINAHL Plus, and Web of Science, as well as grey literature sources will be searched from inception up to December 2022. We will include randomised controlled trials, quasi-experimental and interventional studies of interventions, which aim to improve perinatal outcomes in any HIC. There will be no language restrictions. We will exclude studies presenting only qualitative outcomes and those including mixed populations of migrant and non-migrant women. Screening and data extraction will be completed by two independent reviewers and risk of bias will be assessed using the Quality Assessment Tool for Quantitative Studies. If a collection of suitably comparable outcomes is retrieved, we will perform meta-analysis applying a random effects model. Presentation of results will comply with guidelines in the Cochrane Handbook of Systematic Reviews of Interventions and the PRISMA statement.

**Ethics and dissemination** Ethical approval is not required. Results will be submitted for peer-reviewed publication and presented at national and international conferences. The findings will inform the work of the Lancet Migration European Hub.

## STRENGTHS AND LIMITATIONS OF THIS STUDY

⇒ We will employ rigorous methodology in accordance with the Cochrane Handbook of Systematic Reviews and report in accordance with the Preferred Reporting Items for Systematic Reviews and Meta-Analyses Protocol statement.

⇒ We aim to assess if included studies employed co-production methods at any point in intervention development or analysis.

⇒ The search strategy was developed with an experienced medical librarian and adapted for each database searched.

⇒ No language restrictions will be employed, and we are doing an extensive and systematic grey literature search which is often omitted in similar reviews.

⇒ The certainty of evidence may be limited by the number of studies available and because some studies may be of low quality without a quantitative outcome assessment.

**PROSPERO registration number** CRD42022380678.

## INTRODUCTION
### Rationale

There are an estimated 281 million international migrants in the world, constituting 3.6% of the global population.[1] Of these, approximately 13% are forced migrants, including refugees and asylum seekers who often face significant hardships during migration and when settled in their host country.[2] The remainder are economic migrants who often choose to migrate to reunite with family or for better job prospects, but they may also experience marginalisation due to poverty, social isolation, and discrimination.[1] Between 2000 and 2022, the international migrant population increased by 108 million, and although the majority of international migrants originate from low-income and middle-income countries



(LMICs), increasing numbers hope to settle in high-income countries (HICs).[1] HICs are not always well adapted to care for the needs of marginalised migrants, and many HICs struggle to adapt to this changing migration landscape.[1]

Women constitute approximately half of the international migrant population.[2] Women who migrate and who are pregnant or postpartum can face significant barriers to accessing maternity care, putting them at increased risk of poor mental and physical health.[3–6] They may experience poor social support in their host country, an inability to access to healthcare and language barriers.[5 7] Forced migrant women who are pregnant or postpartum experience additional challenges including premigration stressors such as the trauma of war, transition stressors including dangerous migration journeys and gender-based violence, and postmigration stressors such as poor access to legal entitlements, discrimination and sociocultural barriers in obstetric care.[7 8] They are also more likely to have poor access to antenatal care and experience higher rates of perinatal mortality, miscarriage, and stillbirth than non-refugee women.[9] HICs often have some of the resources needed to support marginalised pregnant and postpartum migrant women, but often struggle to provide optimum care.[8]

To our knowledge, no previous review has sought to quantitatively synthesise the literature on the most effective interventions to improve perinatal outcomes for migrant women in HICs. Balaam *et al* conducted a systematic review in 2020 which aimed to identify social support interventions for asylum-seeking and refugee women in Europe.[10] The findings were qualitatively synthesised and women valued peer support and interventions that addressed their needs in a holistic way.[10] The UK National Institute of Health and Care Research commissioned a systematic review to explore interventions to improve maternity care for migrant women in the UK in 2017.[11] The review included only UK studies and identified just four interventions. These included peer support and specialist maternity care interventions, but they were not robustly evaluated, so it was difficult to draw conclusions on their effectiveness.[11]

### Objectives

This systematic review aims to identify the most effective interventions to improve perinatal outcomes for migrant women in HICs by quantitatively synthesising the literature. The primary objective will be to determine what interventions exist to improve perinatal outcomes for migrant women in HICs. The secondary objective will be to explore the effectiveness of these interventions by exploring the impact on perinatal outcomes. The main outcomes of interest will be rates of preterm birth, birth weight, and number of antenatal or postnatal appointments attended.

### METHODS AND ANALYSIS
### Registration and protocol adherence

This systematic review will be reported in accordance with the Preferred Reporting Items for Systematic Reviews and Meta-Analyses (PRISMA) guidelines and was registered on PROSPERO: CRD42022380678 on 9 December 2022.

### Definitions

For the purposes of this review, migrant women will be defined as being aged 16 years or older and who were born outside their host country. Eligible maternity care interventions are any hospital-based or community-based initiatives offered in the antenatal, perinatal, or postnatal period up to 1-year postpartum. The perinatal period is defined as pregnancy and up to 1-year postpartum.

### Eligibility criteria and patient, intervention, comparison and outcome framework
### Patient, intervention, comparison and outcome framework

► Population: Perinatal migrant women (those who were not born in their host country) aged 16 years or older and living in HICs (defined as being in the World Bank high-income economy category).[12]
► Intervention: Any hospital- or community-based activity undertaken with the aim of improving perinatal outcomes and delivered during the antenatal period and up to 1-year postpartum.
► Control: Usual care if data are available.
► Outcome: The main outcomes of interest will be rates of preterm birth, birth weight and number of antenatal or postnatal appointments attended as these are crucial measures of quality of maternity care in accordance with the WHO guidelines for Quality of Care for Pregnant Women and Newborns.[13] Improvements in perinatal outcomes (rates of miscarriage, preterm birth, stillbirth, birth weight, mode of delivery, APGAR score, maternal/neonatal critical care admission, breastfeeding initiation and duration, maternal/neonatal death, perinatal mental illness); number of antenatal or postnatal appointments attended or change in maternal well-being as assessed by validated mental illness or well-being screening scales, as well as any other outcomes retrieved from included studies.

### Inclusion and exclusion criteria

Observational, quasi-experimental, and experimental intervention studies published from inception will be included. Abstracts, non-empirical research, opinion or editorial pieces will be excluded. If duplicate reports or publications of the same data are retrieved, the less complete or recent version will be excluded if the same data are reported. Studies including only a qualitative outcome assessment will be excluded. Interventions that were not specifically designed or adapted for migrant women in the perinatal period will be excluded. This is to ensure our results are focused on interventions that could be directly implemented for migrant women and have a direct impact on their outcomes. Additionally, we felt this ensures the systematic review is focused and won't return an unmanageable number of results.

## Patient and public involvement

Migrant women who are residing in the UK helped with the review's inception and design, and will also help with data extraction, analysis, and interpretation. They will also help with disseminating the work through co-authorship on peer-reviewed manuscripts and presentations at conferences.

## Search strategy and data sources

EMBASE, EMCARE, MEDLINE and PsycINFO via Ovid, Cochrane Central Register of Controlled Trials (CENTRAL) via Cochrane Library, Scopus, CINAHL Plus via EBSCOHost, and Web of Science from inception to December 2022 (online supplemental appendix 1). Grey literature sources including Google Scholar and trial registries were searched up until December 2022. The first 150 results from the following supplementary sources will be searched: Google Scholar, WHO International Clinical Trials Registry Platform (ICTRP), ClinicalTrials.gov, the WHO Website, and the UN Refugee Agency website. Forward citation searching will be employed on all included articles, and the reference lists of all included articles will also be searched. No language restrictions will be employed. If studies are published in a language other than English, one of the research team who is fluent in that language will assess for likely relevance and extract the data, if appropriate. The article will also be translated using Google Translate and a second reviewer will review its relevance and extract the data, if appropriate. If none of the research team are fluent in the language, we will pay UCL's graduate linguistics programme to translate the article. The Boolean operators 'AND' and 'OR' will be employed to combine the descriptors. An experienced medical librarian helped to develop the search strategy and it has been adapted for each database. A search strategy is provided in online supplemental appendix 1. EndNote will be used to collect and manage studies retrieved.[14] Covidence will be used for deduplication and for study selection.[15]

## Data extraction (selection and coding)

Two independent reviewers will screen the titles and abstracts of all the citations for relevance. Full text manuscripts for relevant articles will be obtained. Full texts will be independently assessed for eligibility using a checklist of the inclusion and exclusion criteria by two independent reviewers. Studies meeting the inclusion criteria will be selected for inclusion in the review. All excluded articles from the full text screening will be retained with reason for exclusion noted. Disagreements between reviewers will be discussed and agreement sought from a third reviewer if necessary. If data are not accessible from the paper, the authors will be contacted. All extracted data will be recorded on the piloted data extraction form by two separate reviewers and cross-checked. The main data fields will be: (A) author, publication year; (B) country; (C) study design; (D) population and baseline characteristics; (E) context (community or hospital based or online); (F) intervention details; (G) control or comparison; (H) timing of outcome measurements; (I) outcome measures (type, scale used, scale validation status); (J) outcomes; (K) quality assessment; (L) reported according to (Consolidated Standards of Reporting Trials) CONSORT guidelines and (M) coproduction methods used. A flow chart will summarise the selection process in line with the PRISMA 2020 guidelines. Study characteristics will be summarised and presented in tables.

## Quality assessment and risk of bias

Two reviewers will perform the critical appraisal independently and this will be independently cross-checked. The Quality Assessment Tool for Quantitative Studies will be used to assess rigour for each included study.[16] This was chosen as it has been developed and validated to assess both observational and experimental studies and shows reliability and validity. It assesses selection bias, study design, confounders, blinding, data collection, withdrawals, intervention integrity and statistical analysis. To assess time-related biases, we will include components of the Risk of Bias in Non-Randomised Studies of Interventions tool.[17] If meta-analysis is conducted, publication bias will be assessed using a funnel plot.

## Data synthesis

Study characteristics including outcomes, for example, raw proportions, mean scores and ORs/risk ratios will be extracted and presented in tabular form. Narrative synthesis will be conducted according to Cochrane guidance and will include the creation of categories of interventions based on included papers, for example, specialist midwifery services, interpreting services and use of a doula.[18 19] If appropriate, heterogeneity between studies will be explored using the $I^2$ statistic. Random effects meta-analysis will be conducted with 95% CIs to allow for expected heterogeneity between different study populations, if appropriate. If possible, pooled estimates of OR/risk ratios with 95% CIs will be calculated to explore outcomes among migrant women compared with usual care. Subgroup analyses according to migration status (economic and forced) and study context (hospital based or community based) will be conducted, if appropriate. Sensitivity analysis according to study quality and method of recruitment will be undertaken. If meta-analysis is not possible due to a lack of standardised outcome data, alternative quantitative methods will be used following the Cochrane Handbook for Systematic Reviews of Interventions guidance.[19] The handbook details several approaches, but it is likely a Harvest Plot will be most appropriate. A Harvest Plot provides a visual extension of vote counting by categorising studies based on their effect (eg, 'beneficial effect' or 'detrimental effect').[20] Vote counting is recommended when there are inconsistent effect measures across studies. Traditional vote counting methods using statistical significance, magnitude of effect or subjective rules have been shown to be misleading.[19] Instead, we will create a standardised binary metric,

'beneficial' or 'detrimental' based on direction of effect for each intervention according to outcome category (eg, birth weight, stillbirth rates or antenatal appointment attendance).[19] We will calculate the proportion of beneficial studies, 95% CI (binomial exact calculation) and p value (binomial probability test) to demonstrate if there is any evidence of an effect. We will present the findings on a Harvest Plot, which will take the form of a 'supermatrix' which visually displays the vote counting results for each intervention.[19] All statistical analyses will be conducted using R Studio.[21] Meta-biases such as publication bias across studies will not be assessed.

## Confidence in cumulative evidence

Confidence in the strength of evidence found will be assessed using the overall assessment outlined in the Quality Assessment Tool for Quantitative Studies checklist.[16] This will be done by two reviewers and possible disagreement will be assessed by third reviewer.

## ETHICS AND DISSEMINATION

Ethical approval is not required for this study as no primary data are being collected. We intend to publish the results in a peer-reviewed open access publication and present results at national and international conferences. The findings will inform the work of the Lancet Migration European Regional Hub.

## DISCUSSION

This systematic review will provide a comprehensive overview of interventions being used to improve perinatal outcomes for migrant women in HICs. It will make a systematic assessment of the most effective interventions, which will help to inform policy and clinical decision-making across these regions. The findings can also be used by researchers planning or adapting maternity care interventions for migrant women in HICs, as well as to guide resource allocation decisions by funders and healthcare managers. Potential limitations include the retrieval of low-quality studies with poor evaluation of outcomes which will inhibit our ability to robustly assess the evidence of effectiveness of these interventions. Attempts to retrieve all available literature have been used such as involving a medical librarian in the development of our search strategy to ensure a broad and sensitive selection of search terms and searching multiple databases and additional supplementary and grey literature sources, however, it is possible that some relevant studies/ data may be missed. The certainty of evidence may also be limited by few studies including quantitative outcome assessments of interventions. Additionally, we are only including studies conducted in HICs which means we may miss effective interventions which were developed and tested in LMICs. Studies including antenatal interventions and subsequent perinatal outcomes can be affected by time-related biases if women are recruited at various points throughout their pregnancy, that is, person-time of observation is not properly accounted for in the design or analysis of a study.[22] We are unable to adjust for these biases in quantitatively synthesising our findings, but will consider the implications of this in our conclusions. We are taking a coproduction approach to the planning, conducting and interpretation of this work by ensuring migrant women who have given birth or been pregnant in their host country are involved throughout. We will also be assessing if included studies have taken a coproduction approach at any point in intervention development, implementation or evaluation to gain an understanding of the prevalence of coproduction in this field. We will identify gaps in the evidence to guide future research priorities.

## Author affiliations

[1]Institute of Health Informatics, University College London, London, UK
[2]Warwick Medical School, University of Warwick, Coventry, UK
[3]Library Services, University College London, London, UK
[4]Women and Children's Health, King's College London, London, UK
[5]Health Department of Public Health, University of Copenhagen, Kobenhavn, Denmark
[6]National Perinatal Epidemiology Unit, Nuffield Department of Population Health, University of Oxford, Oxford, UK
[7]Collaborative Centre for Inclusion Health, Institute of Epidemiology & Health Care, University College London, London, UK
[8]Institute of Epidemiology & Health Care, University College London, London, UK

**Contributors** KS, KO, FAS, MK and RWA conceived of the review. KS, KO, SE, WGH, HR-J, MM, MM-C, GF, SL, FAS, MK and RWA developed the approach and protocol. KS developed the search strategy. KS wrote the first draft of the protocol, and all authors were invited to comment. KS is the guarantor of the review. All authors approved the final version of the protocol.

**Competing interests** None declared.

**Patient and public involvement** Patients and/or the public were involved in the design, or conduct, or reporting, or dissemination plans of this research. Refer to the Methods section for further details.

**Patient consent for publication** Not applicable.

**Provenance and peer review** Not commissioned; externally peer reviewed.

**ORCID iDs**
Kerrie Stevenson http://orcid.org/0000-0001-5881-1402
Hannah Rayment-Jones http://orcid.org/0000-0002-3027-8025
Robert W Aldridge http://orcid.org/0000-0003-0542-0816

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
