## [Reviewer comments · BMJ Open]

ARTICLE DETAILS

TITLE (PROVISIONAL)	Interventions to improve perinatal outcomes amongst migrant women in high-income countries: a systematic review protocol
AUTHORS	Stevenson, Kerrie; Ogunlana, K; Edwards, Samuel; Henderson, William; Rayment-Jones, Hannah; McGranahan, Majel; Marti-Castaner, Maria; Fellmeth, Gracia; Luchenski, Serena; Stevenson, Fiona; Knight, Marian; Aldridge, Robert

VERSION 1 – REVIEW

REVIEWER	Azar Mehrabadi Dalhousie University, Department of Obstetrics & Gynaecology / Department of Pediatrics
REVIEW RETURNED	07-May-2023

GENERAL COMMENTS	This systematic review protocol clearly addresses an important topic. There are persistent inequalities in maternal health outcomes and undertaking a review summarizing interventions to improve perinatal outcomes among migrant women is a worthwhile contribution. The authors state that a previous UK study from 2017 yielded four studies, but that it did not adequately evaluate the included studies. In addition, the previous review did not include high-income countries more broadly. As a result, this proposed study, which has a wider scope, can potentially expand on the previous findings. The methods are thorough and clear. The following are suggestions for improving the protocol. -The study objective(s) could be clarified. The aim “to explore” is somewhat vague, and in the abstract and background (e.g., p.3, line 18-2), the authors refer to a primary and secondary “outcome”. However, it seems that the authors intended to state “objective” rather than “outcome” and the study questions do not quite follow the PICO format which is stated later on. As mentioned, it’s not clear that an intervention to improve health outcomes would be called an outcome. The PICO framework, however, on p. 7. is clear and the objectives/study questions should correspond to this.-On page 8, the authors state they have made the decision to exclude studies with migrant and non-migrant women. This decision should be further justified. This may exclude certain studies that had interventions in pregnancy or postpartum for a wider target population, and that stratified by migrant versus non-migrant status. It’s not clear why studies that assessed more universally targeted interventions, but presented them by migrant status, would not also yield policy relevant findings. Migrant women could potentially benefit to a greater extent from some universal programming (e.g.
---

	housing support, subsidized doula services, etc). -Studies that assess antepartum interventions and the outcomes of preterm birth and related outcome when using a fixed exposure definition can be affected by immortal time bias. The authors may wish to assess whether this bias is evident in the studies they find by using a similar approach described by Ukah et al (https://onlinelibrary.wiley.com/doi/full/10.1002/pds.5504). Otherwise, some interventions may appear beneficial simply due to this bias. Some minor issues: -P.5, line: 21-22: Clarify the sentence: originate from and within. -p.8. line 18: I was not sure why it was stated that grey literature was searched in December 2022. Perhaps this is a typo? -There were few potential study limitations cited.
--	---

REVIEWER	Elena Riza National and Kapodistrian University of Athens, Hygiene, Epidemiology & Medical Statistics
REVIEW RETURNED	05-Jun-2023

GENERAL COMMENTS	This is a very important topic in the promotion of sexual and reproductive health of migrant women and hopefully there will be enough information to inform the review to allow for future suggestions.
---

VERSION 1 – AUTHOR RESPONSE

REVIEWER #1		
The study objective(s) could be clarified. The aim “to explore” is somewhat vague, and in the abstract and background (e.g., p.3, line 18-2), the authors refer to a primary and secondary “outcome”. However, it seems that the authors intended to state “objective” rather than “outcome” and the study questions do not quite follow the PICO format which is stated later on. As mentioned, it’s not clear that an intervention to improve health outcomes would be called an outcome. The PICO framework, however, on p. 7. is clear and the objectives/study questions should correspond to this.	Thank you for this comment. We have edited the objective sections in both the abstract and the introductory section. Abstract: ‘The primary objective of this systematic review is to provide an assessment of what interventions exist to improve perinatal outcomes in HICs. The secondary objective is to assess the most effective interventions for improving perinatal outcomes in HICs.’ Background: ‘Objectives This systematic review aims to identify the most effective interventions to improve perinatal	Page 2-3, Page 5

	outcomes for migrant women in HICs by quantitatively synthesising the literature. The primary objective will be to determine what interventions exist to improve perinatal outcomes for migrant women in HICs. The secondary objective will be to explore the effectiveness of these interventions by exploring the impact on perinatal outcomes. The main outcomes of interest will be rates of pre-term birth, birthweight, and number of antenatal or postnatal appointments attended.'	
On page 8, the authors state they have made the decision to exclude studies with migrant and non-migrant women. This decision should be further justified. This may exclude certain studies that had interventions in pregnancy or postpartum for a wider target population, and that stratified by migrant versus non-migrant status. It's not clear why studies that assessed more universally targeted interventions, but presented them by migrant status, would not also yield policy relevant findings. Migrant women could potentially benefit to a greater	Thank you for this comment. We agree that universally targeted interventions can have an impact on outcomes for migrant women. However, we chose to exclude interventions that weren't specifically designed for migrant women to ensure our results were focused on interventions that could be directly implemented for migrant women and had direct impact their outcomes. Additionally, we felt this ensured the	Page 7
extent from some universal programming (e.g. housing support, subsidized doula services, etc).	systematic review was focused and wouldn't return an unmanageable number of results. We have clarified this further in the methods section. 'Interventions that were	

	not specifically designed or adapted for migrant women in the perinatal period will be excluded. This is to ensure our results are focused on interventions that could be directly implemented for migrant women and have direct impact their outcomes. Additionally, we felt this ensures the systematic review is focused and won't return an unmanageable number of results.'	
Studies that assess antepartum interventions and the outcomes of preterm birth and related outcome when using a fixed exposure definition can be affected by immortal time bias. The authors may wish to assess whether this bias is evident in the studies they find by using a similar approach described by Ukah et al (https://onlinelibrary.wiley.com/doi/full/10.1002/pds.5504). Otherwise, some interventions may appear beneficial simply due to this bias.	Thank you for this comment. We have reviewed our preliminary findings and agree that some of studies may be subject to time- related biases due to varying exposure times in pregnancy. However, some recruited women at the same point in gestation and so will be less likely to be biased in this way. We are unable to address these biases in our analytical approaches as we are unlikely to be carrying out quantitative synthesis, but we have added a discussion of the implications of time-related bias in the discussion. 'Studies including antenatal interventions and subsequent perinatal outcomes can be affected by time-related biases if women are recruited at various points throughout	Page 11

	their pregnancy, i.e. person-time of observation is not properly accounted for in the design or analysis of a study. We are unable to adjust for these biases in synthesising our findings, but will consider the implications of this in our conclusions.'	
P.5, line: 21-22: Clarify the sentence: originate from and within.	Thank you for this comment. This was trying to highlight that most migrants move from low- and middle-income countries, or migrate within them, but we have amended the sentence to make this clearer. 'Between 2000 and 2022 the international migrant population increased by 108 million, and although the majority of international migrants originate from low- and middle-income countries (LMICs), increasing numbers hope to settle in high-income countries (HICs).¹	Page 4
P.8. line 18: I was not sure why it was stated that grey literature was searched in December 2022. Perhaps this is a typo?	Thank you for this comment. This explains that the grey literature and associated websites were searched up until December 2022. We have amended the sentence to make this clearer. 'Grey literature sources including Google Scholar and trial registries were searched up until December 2022'.	Page 7
There were few potential study limitations cited.	Thank you for this	Page 11

	comment. We have added further limitations to the discussions section. 'Potential limitations include the retrieval of low- quality studies with poor evaluation of outcomes which will inhibit our ability to robustly assess the evidence of effectiveness of these interventions. Attempts to retrieve all available literature have been utilised such as involving a medical librarian in the development of our search strategy to ensure a broad and sensitive selection of search terms and searching multiple databases and additional supplementary and grey literature sources, however it is possible that some relevant studies/data may be missed. The certainty of evidence may also be limited by few studies including quantitative outcome assessments of interventions. Additionally, we are only including studies conducted in HICs which means we may miss effective interventions which were developed and tested in LMICs. Studies including antenatal interventions and subsequent perinatal outcomes can be affected by time-related biases if women are recruited at various points throughout their	
--	--	--

	pregnancy, i.e. person-time of observation is not properly accounted for in the design or analysis of a study.²¹ We are unable to adjust for these biases in quantitatively synthesising our findings, but will consider the implications of this in our conclusions.'	
--	---	--

VERSION 2 – REVIEW

REVIEWER	Azar Mehrabadz Dalhousie University, Department of Obstetrics & Gynaecology / Department of Pediatrics
REVIEW RETURNED	11-Jul-2023

GENERAL COMMENTS	I suggest a few minor revisions. The objectives could still be revised so that the objectives in the abstract, introduction and PICO statement match. The abstract objective of "to provide an assessment" is not as clear and is different than the objective in the introduction: "to determine what objectives exist". I am confused why the first outcomes listed in the PICO statement (APGAR maternal/neonatal critical care admission etc.) are not the main outcomes. Are these secondary outcomes then? If so, this could be clarified. The authors use many proxy health outcomes (birthweight, number of appointments) as main outcomes of interest which they argue are mediators in longer term infant health, although this is debatable and the limitations of such outcomes are increasingly being recognized (https://pubmed.ncbi.nlm.nih.gov/34661814/). Using health outcomes that are not proxies is always preferable. Since the first objective is to list outcomes that exist in the literature, unless I have misunderstood, the authors may choose to use more objective markers of health as outcomes where they are available (e.g., severe maternal and neonatal morbidities). In addition, rather than merely discussing time-related biases in the conclusion, the authors could include assessing each study for potential time-related biases in the methods. This would be helpful for readers who could then assess the synthesized findings based on whether or not the authors identified such biases.
---

VERSION 2 – AUTHOR RESPONSE

Reviewer #1 Comments	Author Response & Changes Made	Page in Revised Paper where the Change can be Found
The objectives could still be revised so that the objectives in the abstract, introduction and	Thank you for this comment. We agree the	Abstract & Page 5

PICO statement match. The abstract objective of "to provide an assessment" is not as clear and is different than the objective in the introduction: "to determine what objectives exist".	objectives should be clearer, and have taken your advice by ensuring they match in the abstract, introduction, and the PICO statement (addressed in your following comment). Please see below: Abstract: 'The primary objective will be to determine what interventions exist to improve perinatal outcomes for migrant women in HICs. The secondary objective will be to explore the effectiveness of these interventions by exploring the impact on perinatal outcomes.' Introduction: 'The primary objective will be to determine what interventions exist to improve perinatal outcomes for migrant women in HICs. The secondary objective will be to explore the effectiveness of these interventions by exploring the impact on perinatal outcomes.'	
I am confused why the first outcomes listed in the PICO statement (APGAR maternal/neonatal critical care admission etc.) are not the main outcomes. Are these secondary outcomes then? If so, this could be clarified.	Thank you for this suggestion. We have amended the PICO statement to ensure we state the primary outcomes first.	Page 6

	'Outcome: The main outcomes of interest will be rates of pre-term birth, birthweight, and number of antenatal or postnatal appointments attended as these are crucial measures of quality of maternity care in accordance with the World Health Organisation guidelines for Quality of Care for Pregnant Women and Newborns.^{13'}	
The authors use many proxy health outcomes (birthweight, number of appointments) as main outcomes of interest which they argue are mediators in longer term infant health, although this is debatable and the limitations of such outcomes are increasingly being recognized (https://pubmed.ncbi.nlm.nih.gov/34661814/). Using health outcomes that are not proxies is always preferable. Since the first objective is to list outcomes that exist in the literature, unless I have misunderstood, the authors may choose to use more objective markers of health as outcomes where they are available (e.g., severe maternal and neonatal morbidities).	Thank you for highlighting the difficulties of using proxies as measures of longer-term infant health, and for sharing the helpful article by Wilcox and colleagues. You are correct that we will be listing all the outcomes that exist in the literature. As such, we will aim to report these more objective markers where available. As for the primary outcomes, we have decided to continue to prioritise these given the WHO guidelines for Quality of Care for Pregnant Women and Newborns recognise these as crucial measures of quality of maternity care, but have decided to amend the wording to remove the comment about these being	Page 6 & 7

	mediators of longer term infant health. We have also added text to highlight that we will be listing all outcomes in the literature. 'The main outcomes of interest will be rates of pre-term birth, birthweight, and number of antenatal or postnatal appointments attended as these are crucial measures of quality of maternity care in accordance with the World Health Organisation guidelines for Quality of Care for Pregnant Women and Newborns.¹³ Improvements in perinatal outcomes (rates of miscarriage, preterm birth, stillbirth, birthweight, mode of delivery, APGAR score, maternal/neonatal critical care admission, breastfeeding initiation and duration, maternal/neonatal death, perinatal mental illness); number of antenatal or postnatal appointments attended; or change in maternal wellbeing as assessed by validated mental illness or wellbeing screening scales, as well as any other outcomes retrieved from included studies.'	
--	--	--

In addition, rather than merely discussing time- related biases in the conclusion, the authors could include assessing each study for potential time-related biases in the methods. This would be helpful for readers who could then assess the synthesized findings based on whether or not the authors identified such biases.	Thank you for this comment. We have considered again the possibility of assessing time-related biases in our analysis. Our quality assessment tool can be used for both experimental and observational studies, which provides us with flexibility given that many types of studies that are likely to be included. However, it doesn't specifically address time-related biases. As such, we will explicitly assess time- related biases by including components of the Risk of Bias in Non-Randomised Studies (ROBINS- I) tool with the existing tool. We have updated the methods to reflect this change. 'Quality Assessment / Risk of Bias Two reviewers will perform the critical appraisal independently and this will be independently cross-checked. The Quality Assessment Tool for Quantitative Studies will be used to assess rigor for each included study.¹⁶ This was chosen as it has been developed and validated to assess both observational and experimental studies and shows reliability and validity. It assesses selection bias, study design, confounders, blinding, data collection, withdrawals, intervention integrity, and statistical	Page 9
---	--	---------------

	analysis. To assess time-related biases we will include components of the Risk of Bias in Non-Randomised Studies (ROBINS-I) tool.¹⁷ If meta-analysis is conducted, publication bias will be assessed using a funnel plot.'	
--	---	--